# Use of GRP Pipe Waste Powder as a Filler Replacement in Hot-Mix Asphalt

**DOI:** 10.3390/ma13204630

**Published:** 2020-10-16

**Authors:** Ahmet Beycioğlu, Orhan Kaya, Zeynel Baran Yıldırım, Baki Bağrıaçık, Magdalena Dobiszewska, Nihat Morova, Suna Çetin

**Affiliations:** 1Department of Civil Engineering, AAT Science and Technology University, Adana 01250, Turkey; abeycioglu@atu.edu.tr (A.B.); okaya@atu.edu.tr (O.K.); zbyildirim@atu.edu.tr (Z.B.Y.); 2Department of Civil Engineering, Çukurova University, Adana 01250, Turkey; bbagriacik@cu.edu.tr; 3Faculty of Civil and Environmental Engineering and Architecture, UTP University of Science and Technology, 85–796 Bydgoszcz, Poland; 4Department of Civil Engineering, Isparta University of Applied Science, Isparta 32260, Turkey; nihatmorova@isparta.edu.tr; 5Department of Ceramics, Çukurova University, Adana 0125, Turkey; cetins@cu.edu.tr

**Keywords:** hot-mix asphalt, GRP composite pipe, manufacturing waste powder, filler replacement, sustainability

## Abstract

There is an increasing global trend to find sustainable, environmentally friendly and cost-effective materials as an alternative to limited natural raw materials. Similarly, the use of waste materials has been gaining popularity in the production of hot-mix asphalt (HMA). In this study, the sustainable use of glass-fiber-reinforced polyester (GRP) pipe waste powder (GRP-WP), gathered from the cutting and milling process of GRP pipe production, utilizing it in asphalt mixes as a filler, is evaluated based on lab testing to find out: (i) if it produces similar or better performance compared to the most conventionally available filler material (limestone) and, (ii) if so, what would be the optimum GRP-WP filler content to be used in asphalt mixes. For this reason, an experimental test matrix consisting of 45 samples with three different amounts of binder content (4%, 4.5% and 5.0%), and a 5% filler content with five different percentages of the GRP-WP content (0%, 25%, 50%, 75% and 100% replacement by weight of the filler), was prepared to figure out which sample would produce the similar Marshall stability and flow values compared to the control samples while also satisfying specification limits. It was found that the samples with 4.5% binder content, 3.75% GRP-WP and 1.25% limestone filler content produced the results both satisfying the specification requirements and providing an optimum mix design. It is believed that use of GRP-WP waste in HMA production would be a very useful way of recycling GRP-WP.

## 1. Introduction

The World Bank has announced that our global waste production is increasing day by day. By 2030, the world is expected to generate 2.59 billion tons of waste annually. By 2050, the world is expected to increase its waste generation by 70 percent, from 2.01 billion tons of waste in 2016 to 3.40 billion tons of waste annually [1]. Recycling and disposal are two current waste management options [2]. When the current amounts of waste and its potential increase are considered, recycling wastes instead of disposing of them is more important for sustainability.

Asphalt concrete is one of the vital structures in terms of civil engineering and is used in very large-scale applications including roads and waterproofing due to its high resistance to durability, water resistance and good stability properties [3]. The global road network consists of more than 36 million km of unpaved and paved road network. Since this is an engineering field with such a great production potential, highway construction is a dominant industry consuming substantial amounts of natural resources, especially mineral aggregates derived from quarry extraction. In the last few decades, recycling of industrial waste materials in pavement preservation, maintenance and reconstruction has been a very popular way to achieve sustainable solutions. Billions of tons of waste materials have been produced annually around the world, and pavement applications can be one of the best ways to consume these wastes by reducing the accumulation of landfills, saving raw materials extracted from the environment and consuming lower amounts of energy [4,5,6,7,8,9].

As is well known, asphalt concrete is a composite material itself, containing asphalt cement as binder, coarse and fine aggregates and fillers [3]. In the United States and India, powder materials passing through a 0.075 mm sieve are defined as fillers, while in Europe, powder materials passing through a 0.063 mm sieve are classified as fillers [9]. Despite being used in limited concentrations, the inclusion of fillers in asphalt mix has significant influences on the properties of asphalt mixes. As a result of a comprehensive literature review, Choudhary et al. [9] summarized the benefits of using fillers in the asphalt mix as follows: satisfying the aggregate gradation specification and influencing the strength and volumetric requirements of the mix; reducing optimum bitumen content and material cost of the mix; stiffening bitumen to improve the mechanical properties of the mix and increasing not only the ability of mixes to resist permanent deformation at high temperatures but also cracking resistance at low temperatures and fatigue life at intermediate temperatures; influencing the “bond” in the aggregate–bitumen system, which further affects the moisture sensitivity of the mix; slowing down the aging process of asphalt mixes by either catalyzing oxidation or by hindering the diffusion of oxygen in mastic; influencing the thermal performance of asphalt mixes; affecting the constructability of a mix by influencing its mixing and compaction temperature. Hence, the choice of suitable filler is a primary concern amongst field engineers [10,11,12,13,14,15,16,17,18,19,20,21,22].

The conventional mineral fillers are directly supplied by mining from natural resources. The filler is a material that is consumed continuously as it is one of the basic materials used in asphalt road construction. This continuous use may even cause difficulties in material supply in some regions, leading to the imposition of restrictions on mining in several regions and reducing the availability of good-quality aggregates at shorter haul distances. If this situation is evaluated in terms of the overall cost of projects, providing the aggregates from longer distances increases their transportation cost as well as the overall cost of pavement constructions. Recycling the waste materials as fillers by using them in place of conventional fillers looks very effective for sustainability practices in pavement construction.

The concept of sustainability is very important and no natural materials are unlimited. For this reason, it will always be desirable to find affordable and environmentally friendly alternative materials in asphalt mixtures. Moreover, studies on the use of waste materials in asphalt mixes will be very valuable to meet the demands of the various agencies’ environmental rating systems (such as LEED and BREEAM, which are the construction project certification programs that signify a certain level of environmentally friendly design achievement of a building) [4]. The current literature shows that the use of waste materials in pavement designs has been studied for many years. Especially, the waste materials used as fillers include tire-derived fuel fly ash [23], rice husk ash [24,25], recycled waste lime [26], andesite waste [27], fly ash [28], red mud waste [29], construction and demolition waste, brick powder [30], Kota Stone [31], bauxite residue [32] and waste glass powder [33], etc. The reutilization of the fine glass-fiber-reinforced polyester (GRP) powder as a partial cement replacement, partial fine aggregate replacement and filler addition for self-compacting concrete, as well as its influence on the durability of the cementitious products, has already been widely exploited [34,35,36,37,38,39,40,41]. However, there are not many studies found in the literature investigating the use of GRP waste as a filler in asphalt mixes [42].

GRP using thermosetting resins is increasingly utilized for a wide range of applications, such as transportation, construction and energy. Indeed, the diversity in the manufacturing, possibilities ranging from unidirectional laminates to randomly oriented fiber compounds and their attractive mechanical properties make these materials very appealing [43]. The GRP industry by way of the manufacturing process produces considerable quantities of waste materials such as fibers, polymers, particle pipe powder, etc., and to landfill such solid wastes would cause a major environmental hazard.

According to the literature, there is no clear statistical information on the amount of pipe powder that emerged in GRP pipe manufacturing. The GRP factory which supported this research reported that the amounts of pipe powder (PP) produced in its manufacturing were 31.89 tons in 2015, 41.22 tons in 2016, 73.1 in 2017, 134.42 tons in 2018 and 1333.62 tons until October in 2019.

One of the biggest problems in the GRP pipe industry is waste recycling. The variety of solutions for recycling GRP manufacturing wastes with effective technologies is increasing as landfilling these wastes has negative environmental impacts [44,45,46]. Reducing the waste by mechanical, thermal and chemical approaches has been implemented by various industrial sectors [47]. Industrial companies have to focus on industrial scale composite recycling to improve acceptable waste management solutions to meet the expectations of sustainability [48,49].

The waste management of GRP materials, particularly those made with thermosetting resins, is a critical issue for the composites industry because these materials cannot be reprocessed. Therefore, most thermosetting GRP waste is presently sent to landfills, in spite of the significant environmental impact caused by their disposal in this way. The limited GRP waste recycling worldwide is mostly due to its intrinsic thermosetting properties, lack of characterization data and the unavailability of viable recycling and recovery routes.

In this study, sustainable use of glass-fiber-reinforced polyester pipe waste powder (GRP-WP), utilizing them in asphalt mixes as fillers, will be evaluated based on lab testing to find out: (i) if it produces similar or better performance compared to the most conventionally available filler material (limestone) and, (ii) if so, what would be the optimum GRP-WP filler content to be used in asphalt mixes. It is believed that the use of GRP-WP waste in asphalt mixes would be a very useful way of recycling the huge amount of GRP pipe waste powders. Considering the existence of a wide variety of industrial sectors and the potential hazards posed by the wastes, the potential use of waste in hot-mix asphalt or other engineering materials should continue to be explored for many years to come.

## 2. Materials and Methods

### 2.1. Materials 

In this study, crushed limestone aggregates obtained from a quarry located in the southern region of Turkey, namely from Adana Province, that have been commonly used in asphalt pavement applications, were used. In terms of gradation of the mix, a coarse aggregate of 58%, passing between 25–4.75 mm sieves, a fine aggregate of 37%, passing between 4.75–0.075 mm sieves, and a filler of 5% were used. Sieve analysis results of the mix was within the limits specified by Turkish Highway Construction Specifications (HTS) [50]. Table 1 shows sieve analysis results of the mix used in this study as well as the specification limits.

Physical, mechanical and durability properties of the aggregates used in this study were also determined based on American (ASTM) standards (Table 2).

As part of this study, along with limestone-based coarse and fine aggregates, GRP-WP (Figure 1), a waste material produced during the cutting and milling of GRP pipes, was used as a replacement for filler in different ratios. In order to use GRP-WP as filler, GRP-WP was first sieved through a 0.075 mm sieve and the powder passing through the sieve was used as filler in asphalt mixes.

GRP-WP and its scanning electron microscopy (SEM) image are shown in Figure 2. As can be seen in Figure 2, the SEM image reveals that GRP-WP contains a considerable amount of micro-sized chopped glass fibers (CGF). Considering these micro-CGFs, one of the motivations of this study was that GRP-WP would be a good candidate to be used as filler, potentially improving the performance of asphalt concretes.

To prepare the Marshall samples, an asphalt binder with a 50–70 penetration grade (found as 58 in this study) was used. The physical properties of this binder are provided in Table 3.

### 2.2. Methodology

In this study, asphalt concrete samples were prepared by using various amounts of binder content (3.5%, 4%, 4.5%, 5% and 5.5%) and 5% filler content, based on the limits specified by HTS [49]. For each binder content amount, three samples were prepared, making a total of 15 samples (3 × 5). All the produced samples were tested based on Marshall stability (MS) test (ASTM D 6927) and stability, flow as well as bulk specific gravity (Gsb), air content (Va), voids in mineral aggregate (VMA) and voids filled with asphalt (VFA) results were obtained so that optimum binder content could be determined (Figure 3a–f).

The optimum binder content for the mix design was determined by taking the average value of the four binder content amounts obtained based on the following criteria and the graphs shown in Figure 3.
Binder content corresponding to the maximum stability (found as 4.64 for MS = 1143)Binder content corresponding to the maximum bulk specific gravity (found as 4.65 for Gsb = 2409).Binder content corresponding to the median value of the specification design limits of air content in the total mix (found as 4.21 for Va = 5%) [50].Binder content corresponding to the median value of the specification design limits of VFA in the total mix (found as 4.51 for VFA = 67.5%) [50].

Based on the criteria above, the optimum binder content was calculated as:(1)$$\frac{\left(4.64+4.65+4.21+4.51\right)}{4}=4.45\%$$

Thereafter, an experimental test matrix consisting of 45 samples with three different amounts of binder content (4%, 4.5% and 5.0%) and a 5% filler content with five different amounts of GRP-WP content (0%, 25%, 50%, 75% and 100% filler replacement) was prepared to determine which sample would produce the highest Marshall stability and flow values as well as satisfying specification limits (Table 4). As can be seen in Table 4, the control sample contained only limestone as filler while other samples contained GRP-WP in different ratios as a replacement for limestone filler. The samples containing GRP-WP were named pipe powder asphalt concrete (PPAC) (Table 4).

## 3. Results

In this part of the paper, 45 Marshall samples explained in Table 4 are tested for Marshall stability (MS), flow, VMA, VFA and Va. Figure 4 shows mean values of the MS results along with the standard deviation distributions for the samples tested. As can be seen in Figure 4, the highest MS value was obtained for the samples with 2.5% GRP-WP and 2.5% limestone (LS) filler content. MS results tend to increase as GRP-PP content increases until a GRP-WP content of 2.5% is reached, and then they start to decrease. Moreover, the highest MS value of 1373.9 kgf was observed for the sample with 4% binder content, 2.5% GRP-WP and 2.5% LS filler content, whereas the lowest MS value of 939.4 kgf was observed for the sample with 4.5% binder content, 5% GRP-WP and no LS filler content. Compared to the control samples, all samples except for the samples with 5% GRP-WP and no LS filler content produced higher MS values.

In Turkey, all state highways under the jurisdiction of the General Directorate of Highways are designed to consist of three stabilized layers (an asphalt wear course at the very top, another asphalt layer (called a binder course) underneath it and an asphalt stabilized base course under the binder course) and a granular subbase course laying on a subgrade. Overall, considering the MS test results, all 45 samples were found to produce adequate MS values to meet the strength requirement needed for binder courses [50].

The flow value is a measure evaluating the behavior of asphalt mixes subjected to traffic loadings and representing the plasticity and elasticity properties of the mixes. Furthermore, the flow value, the vertical deformation value at the maximum load, is a parameter related to the internal friction and cohesion of the compacted asphalt mixes, where it is inversely proportional to the internal friction value [51]. Figure 5 shows mean values of the flow results along with the standard deviation distributions for the samples tested. As can be seen in Figure 5, the lowest flow value was obtained as 1.22 mm for the sample with 4% binder content, 5% GRP-WP and no LS filler content, while the highest flow value was obtained as 2.04 mm for the sample with 4.5% binder content, 3.75% GRP-WP and 1.25% LS filler content. HTS [50] requires asphalt mixes to have at least 2 mm flow value to be used in binder courses. It was observed that the samples with 4.5% and 5% binder content and 3.75% GRP-WP and 1.25 % LS filler content as well as the control samples with 4% and 4.5% binder content produced sufficient flow value to meet the specification limits (Figure 5).

Figure 6 shows mean values of the VMA results along with the standard deviation distributions for the samples tested. As can be seen in Figure 6, VMA results were found to increase as GRP-WP content increased. While VMA results of the control samples were in the range of 12.92% to 14.60%, VMA results of the samples with GRP-WP fillers linearly increased as GRP-WP content increased, reaching its maximum value of 16.27% for the sample with 5% binder content, 5% GRP-WP and no LS filler content. Compared to the control samples, VMA results of the samples with 5% GRP-WP and no LS filler content increased by 19.97%, 18.90% and 11.44% for the binder content amounts of 4%, 4.5% and 5%, respectively.

Figure 7 shows mean values of the VFA results along with the standard deviation distributions for the samples tested. As can be seen in Figure 7, VFA results tend to decrease as GRP-WP content increases. While VFA results of the control samples with 4%, 4.5% and 5% binder content were 61.51%, 67.38% and 68.33%, respectively, they decrease as GRP-WP content increases, reaching its minimum value for the samples with 5% GRP-WP and no LS filler content. VFA results of the samples with 5% GRP-WP and no LS filler content and 4%, 4.5% and 5% binder content were found to be 49.74%, 55.01% and 60.15%, respectively.

Figure 8 shows mean values of the air content (Va) results along with the standard deviation distribution results for the samples tested. As can be seen in Figure 8, Va results tend to increase as GRP-WP content increases. Moreover, Va results of the control samples with 4%, 4.5% and 5% binder content were 4.97%, 4.36% and 4.63%, respectively. It was observed that Va results increase as GRP-WP content increases. However, Va results of the control samples were between those of the samples with 2.5% GRP-WP and 2.5% LS filler content and 3.75% GRP-WP and 1.25% LS filler content. Furthermore, Va results of the samples with 5% GRP-WP and no LS filler content and 4%, 4.5% and 5% binder content were 7.79%, 7.16% and 6.48%, respectively. Samples with the Va results within the limits of the specification were as follows: all control samples and all samples with 3.75% GRP-WP and 1.25% LS filler content. However, no samples with 5% GRP-WP and no LS filler content amounts were found to meet the Va requirements [50].

## 4. Discussion and Conclusions

In this study, the potential use of GRP-WP as a filler was evaluated. First, the optimum binder content was determined for the asphalt mixes with 5% filler content fully composed of LS fillers. The optimum binder content was calculated as 4.5% by taking the average value of corresponding binder content amounts producing maximum values of MS and Gsb as well as median values of specification limits for Va and VFA (Va value of 5% and VFA value of 67.5%). Then, an experimental test matrix consisting of 45 samples with three different amounts of binder content (4%, 4.5% and 5.0%), and a 5% filler content with five different percentages of GRP-WP content (0%, 25%, 50%, 75% and 100% replacement by weight of the filler), was prepared in order to determine which sample would produce the highest Marshall stability while satisfying specification limits for flow and volumetric results. Considering the all test results for MS, flow, VMA, VFA and Va, it was observed that the samples with 4.5% binder content, 3.75% GRP-WP and 1.25% LS filler content produced the results both satisfying the specification requirements and providing an optimum mix design. 

The main objective of this study was to identify if GRP-WP, a waste material, could be used as a filler replacement by evaluating if it produces similar performance compared to the LS filler. In this way, a sustainable, environmentally friendly and cost-effective use of this material could be found. One of the ways to evaluate if a mixture provides similar performance is to compare the MS and flow results of the mixtures with the control samples. MS is measure of strength in asphalt mixtures. Higher values of MS results of the asphalt mixtures are desirable in order to show that they resist shoving and rutting [51]. Therefore, a minimum MS value is specified for the asphalt mixtures, considering the level of traffic the mixture is designed for. In terms of flow results, the maximum allowable flow values in the specifications control the plasticity and maximum allowable binder content, while the lowest flow values control the brittleness and strength of the mixes [51]. Therefore, it is required that the flow results of the asphalt mixtures must be between the lower and upper specification limits. Overall, it was found that the samples prepared with GRP-WP filler replacement produced similar MS and flow values compared to the control samples as well as satisfying the specification limits, except for the case when all LS fillers were replaced with GRP-WP. The samples with 5% GRP-WP and no LS filler produced consistently poorer performance compared to the control samples. One of the main objectives of this study was satisfied in that there was no significant loss of performance of the samples with GRP-WP, except for the case with 5% GRP-WP and no LS filler, compared to the control samples.

In order to identify how GRP-WP filler replacement changes the micro-structure and behavior of the mix, scanning electron microscope (SEM) images of the asphalt mixes were also analyzed. It is known that GRP is a well-designed high-performance composite containing glass fiber, silica sand and polyester resin in its structure [48]. Having these materials in its structure, GRP is a very well-interlocked and tightly bonded material. It is known that when mixed with asphalt binder, filler and asphalt binder creates a filler–asphalt mastic, a high-consistency matrix, cementing larger aggregate particles together and so affecting the mechanical and physical properties of the asphalt mixtures [52]. GRP-WP contains a considerable amount of micro-size chopped glass fibers (CGF), silica sand particles and polyester resin particles on the CGF’s surface. When it is mixed with asphalt binders, due to its fibrous properties, it might have produced a stiff mastic that binds aggregate particles together and produces a mixture with a similar performance as the mixtures with LS fillers. This might explain why the samples with GRP-WP exhibited similar MS and flow values compared to the control samples. 

Numerous studies showed that there are correlations between volumetric properties of the mixes and their performance [53,54]. The durability of an asphalt mixture is highly affected by its Va. Too low Va may lead to flushing, whereas too high Va might cause water damage and rutting in the mixtures. On the other hand, VMA shows the amount of space needed to accommodate the effective volume of binder and the volume of air voids needed in the asphalt mixture. In order to achieve a durable binder film thickness, a minimum VMA value is required [53]. VFA, the total amount of voids between aggregate particles in the compacted mixture that are filled with asphalt binder, is specified to avoid less durable asphalt mixes. It was also shown that a decrease in Va and increase in VFA are some of the ways to reduce the cracking potential of the mixtures [54]. In order to produce a well-performing asphalt mixture, all volumetric properties of the mixes must satisfy corresponding specification limits.

There was a general trend among the samples with GRP-WP that as GRP-WP content increases, VMA and Va results increase but VFA results decrease. Higher Va results with increasing GRP-WP content might be due to the lower level of adsorption and chemical exchange between silica and asphalt, causing more free binder in the mixtures [55]. In other words, as found by some other studies [4], GRP-WP has lower porosity compared to LS. Lower porosity of GRP-WP causes lower absorbance of bitumen in the mixture, leading to more free binder in the mixtures. Therefore, as GRP-WP filler content increases, the amount of free binder in the mixture increases, causing the higher level of Va of the mixes [4,55]. In order to solve this issue, an optimum asphalt binder content could be determined for each GRP-WP filler content to avoid excessive Va.

It was also observed during the compaction of the mixtures that even though the same number and level of Marshall hammer blows were used in all samples, a noticeable decrease in the level of compaction occurred as GRP-WP content increased. Zulkati et al. [13] demonstrated that stiffened asphalt–filler mastic produces a tougher mix that is hard to compact. The role of filler in the asphalt–binder mastic is that fillers act as tiny rollers during the compaction, similar to the role of friction-lubricate agents, given that they have regular and spherical shapes. The asphalt–filler mastic with regular-shaped fillers, such as LS [13], leads to a faster and smoother orientation of the larger aggregates, causing less compaction resistance. As discussed earlier based on Figure 2, the SEM image of the GRP-WP reveals that GRP-WP contains a considerable amount of micro-sized chopped glass fibers (CGF). Due to the irregular shapes of these CGFs, they have significantly longer lengths compared to their widths, and they might have negatively impacted the workability of the asphalt mixtures, causing a reduction in compaction, as discussed by several other studies, such as those presented by Melotti et al. [56] and Wróbel et al. [57]. Resistance to compaction might be one of the main reasons why Va results increase as GRP-WP content increases. A similar trend was observed in some other studies that [58,59] reduction in compaction might cause an increase in Va. 

It was observed in this study that the samples with 5% GRP-WP content with no LS filler exhibited significantly higher levels of Va and VMA as well as significantly lower levels of VFA compared to the control samples. As a result, the same samples produced significantly lower MS and flow results compared to the control samples. This shows that the volumetric properties of the samples are one of the key factors in determining the performance of asphalt mixtures.

Volumetric properties of asphalt mixtures are related to each other. VFA is inversely related to Va and VMA: as Va and VMA increase, the VFA decreases. This is why, as GRP-WP content increases, Va and VMA results increase but VFA results decrease [58]. 

This study demonstrated that GRP-WP, a sustainable material, could be successfully used as a filler replacement, not only satisfying speciation limits but also performing as well as an LS filler when it is used in its optimum content. It is believed that use of GRP-WP waste in asphalt mixes would be a very useful way of recycling the huge amount of GRP pipe waste powders. Therefore, it is imperative that, like any other newly introduced material, a systematic study like this study should be carried out to find out its optimum content and identify if it produces as successful performance as its conventionally available counterpart, such as LS in this case. Considering the existence of a wide variety of industrial sectors and the potential hazards posed by the wastes, the potential use of waste in asphalt concrete or other engineering materials should continue to be explored for many years to come.

## Figures and Tables

**Figure 1 materials-13-04630-f001:**
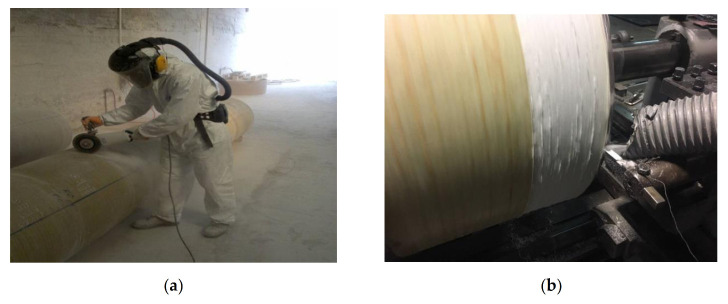
(**a**) Cutting and (**b**) milling process of GRP pipes.

**Figure 2 materials-13-04630-f002:**
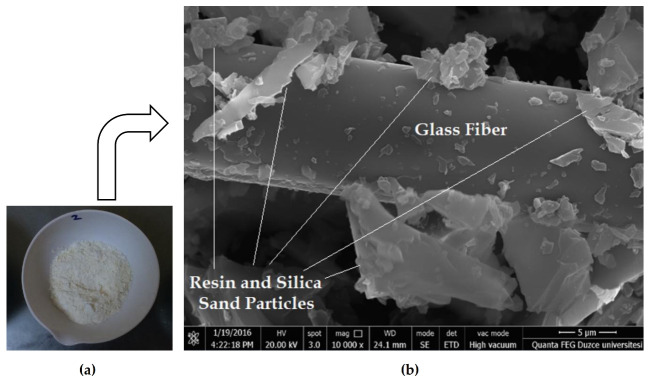
(**a**) GRP-WP and (**b**) SEM image of GRP-WP.

**Figure 3 materials-13-04630-f003:**
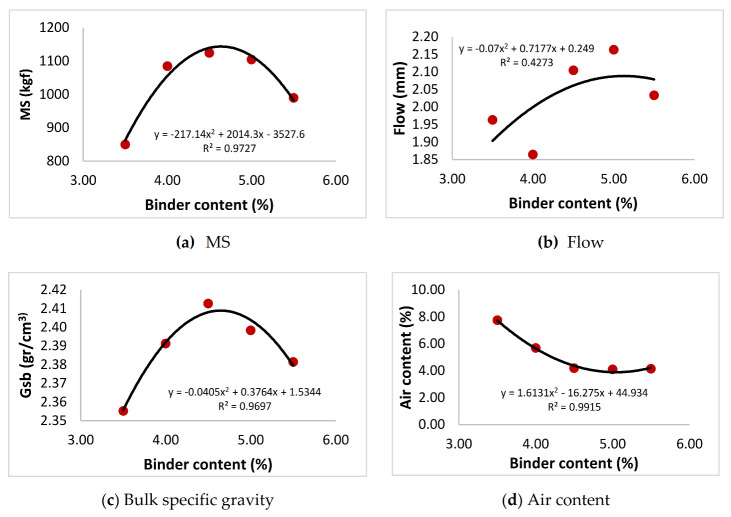
Test results for various binder content values.

**Figure 4 materials-13-04630-f004:**
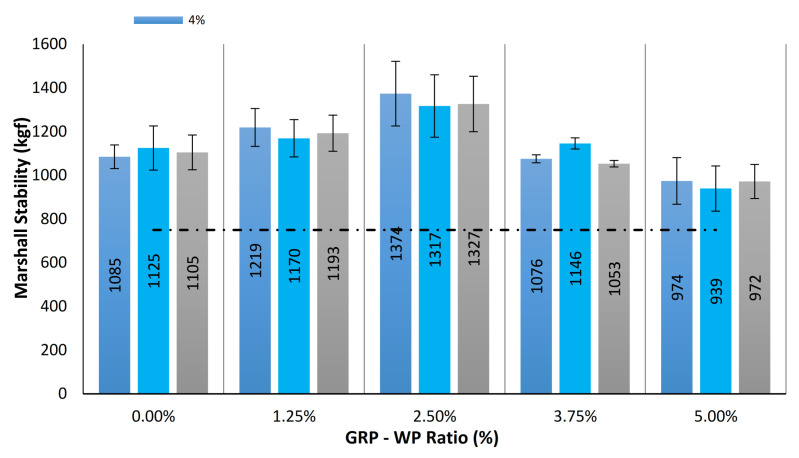
Comparison of MS results for the 45 samples tested.

**Figure 5 materials-13-04630-f005:**
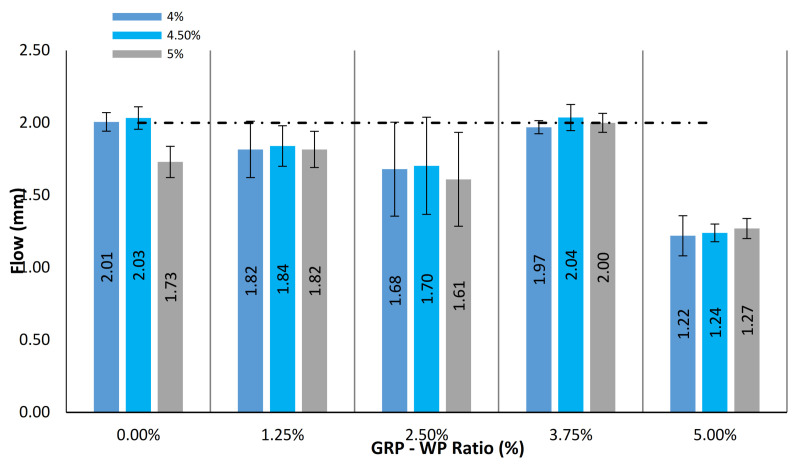
Comparison of flow results for the 45 samples tested.

**Figure 6 materials-13-04630-f006:**
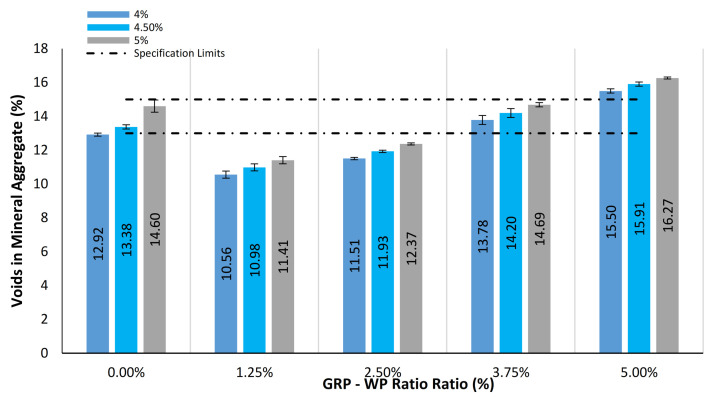
Comparison of VMA results for the 45 samples tested.

**Figure 7 materials-13-04630-f007:**
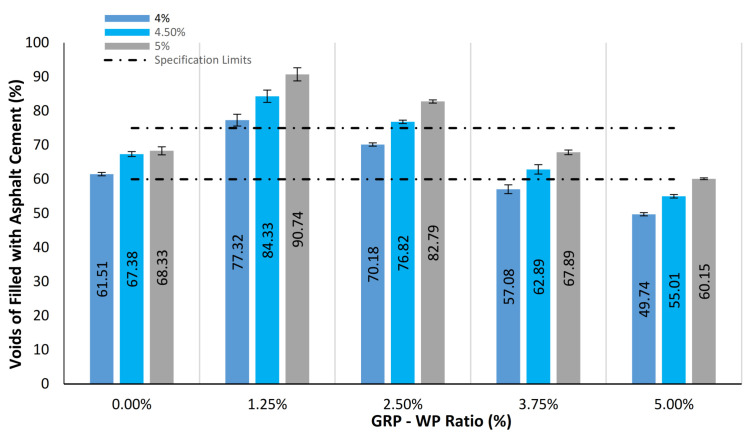
Comparison of VFA results for the 45 samples tested.

**Figure 8 materials-13-04630-f008:**
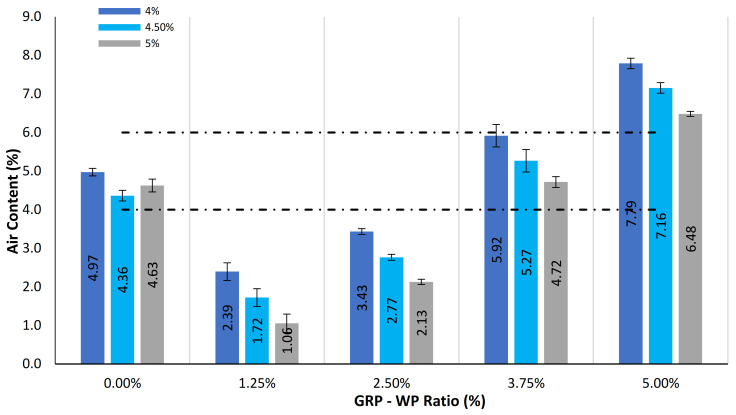
Comparison of air content (Va) results for the 45 samples tested.

**Table 1 materials-13-04630-t001:** Gradation and specification limits.

Sieve Diameter	Limit Values	Gradation of Mixture	Weight (g)
(mm)	(HTS) % Passing [50]	% Passing	
25 mm	100	100	0
19 mm	80–100	91.0	103.0
12.5 mm	58–80	67.5	270.9
9.5 mm	48–70	59.9	86.8
4.75 mm	30–52	42.1	205.5
2.00 mm	20–40	26.0	185.3
0.425 mm	8–22	11.0	172.3
0.180 mm	6–14	7.4	41.4
0.075 mm	2–7	5.0	27.3
Filler	0	0	57.5
Total	100%	100%	1150

**Table 2 materials-13-04630-t002:** Physical, mechanical and durability properties of aggregates.

Properties		Results	Tests Standards
Specific gravity (g/cm^3^)			
Coarse aggregate	Apparent specific gravity	2.771	ASTM C 127
	Bulk specific gravity	2.729	
Fine aggregate	Apparent specific gravity	2.766	ASTM C 128
	Bulk specific gravity	2.646	
Filler	Bulk specific gravity	2.778	ASTM C 128
Dry rodded unit weight (g/cm^3^)		1.914	ASTM C 29
Unit weight (bulk density) (g/cm^3^)		1.725	ASTM C 29
Abrasion loss (%) (Los Angeles)		26.12	ASTM C 131
Flatness index (%)		14.65	ASTM D 4791
Resistance to disintegration by sulfate (weight loss %)		4.67	ASTM C 88

**Table 3 materials-13-04630-t003:** Physical properties of the asphalt binder.

Properties	Results	Test Standards
Source of the Binder	Kırıkkale, Turkey	-
Penetration Grade (25 °C)	58 (50/70)	ASTM D 5
Softening Point (°C)	48.5	ASTM D 36/D 36 M
Specific Gravity (g/cm^3^)	1.040	ASTM D 70–09 e 1
Ductility (25 °C)	>100 cm	ASTM D 113–07
Loss on Heating (%)	0.43	ASTM D 6–95
Flash Point (°C)	280	ASTM D 92–05 a
Viscosity (at 135 °C, cP)	430.23	ASTM D 4402–06
Viscosity (at 165 °C, cP)	120.95	ASTM D 4402–06

**Table 4 materials-13-04630-t004:** Experimental test matrix used in this study.

Sample Name	Aggregates %			Limestone in Filler (%)	GRP-WP in Filler (%)
	Coarse	Fine	Filler		
Control Sample	60	35	5	5	0
PPAC 1	60	35	5	3.75	1.25
PPAC 2	60	35	5	2.5	2.5
PPAC 3	60	35	5	1.25	3.75
PPAC 4	60	35	5	0	5

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
