# Peer review of "Use of GRP Pipe Waste Powder as a Filler Replacement in Hot-Mix Asphalt"

_materials, 2020, doi:10.3390/ma13204630_

Round 1
Reviewer 1 Report
The main areas that ned attention are:
Literature review - need to comment on GRP waste in asphalt application and research(note US patent on this - see comments in attached artilce draft).
Results - data presented needs error/standard deviation including to reveal 'spread' of data.
Discussion - the reason why substitution of LS filler with GRP-WP causes changes to the properties of the asphalt concrete need to be presented and discussed (it is also unlikely that glass fibres in the GRP-WP influence 'strength' of mix as the grinding process will reduce their aspect ratio to much below the critical value for effective 'stiffening/strengthening'.
Please refer to comments in attached annotated manuscript

Author Response
Manuscript ID: materials-935635
Title: Use of GRP Pipe Waste Powder as a Filler Replacement in Hot-Mix Asphalt
Authors: Ahmet BeycioÄŸlu, Orhan Kaya, Zeynel Baran Yıldırım, Baki BaÄŸrıaçık, Magdalena Dobiszewska*, Nihat Morova, Suna Çetin
The authors sincerely appreciate all the constructive comments received from the reviewers for this paper. All of the comments were thoroughly addressed in the revised version of the manuscript. Detailed responses to the reviewer comments are presented below.
REVIEWER 1:
Comment: The main areas that need attention are:
Literature review - need to comment on GRP waste in asphalt application and research(note US patent on this - see comments in attached article draft).
Response: The authors appreciate the reviewer’s comment. US patent (US 2004/0132842 A1) that reviewer mentioned was thoroughly reviewed. The patent was found to be only related to the methods (using what kind of techniques and machinery) that scrap fiberglass could be used in both concrete and asphalt construction. Although it is not directly related to the scope of the paper (the scope of the paper is finding the optimum GRP-WP content to be used as a filler in asphalt mix), the authors decided to cite this US patent in the literature review part of the revised manuscript, as suggested by the reviewer.
The sentence in lines 90-91“However, there is no study found in the literature investigating use of Glass fiber reinforced polyester (GRP) waste as filler in asphalt mixes.” was revised in the revised manuscript as “However, there are not many studies found in the literature investigating use of Glass fiber reinforced polyester (GRP) waste as a filler in asphalt mixes [34].” Reference [34] is for the US patent.
Comment: Results - data presented needs error/standard deviation including to reveal 'spread' of data.
Response: The authors appreciate the reviewer’s comment. As stated in lines 155-, “For each binder content, three samples were prepared, making a total of 15 samples (3x5). All the produced samples were tested based on Marshall stability (MS) test and stability, flow as well as bulk specific gravity (Gsb), air content (Va), voids in mineral aggregate (VMA) and voids filled with asphalt (VFA) results were obtained so that optimum binder content could be determined (Figure 3 a-f).” Results shown in the figures in Results section are based on the mean values of the three samples tested for each case. As requested by the reviewer, the figures were revised in the revised manuscript that both mean values as well as the standard deviation distributions of the results for each test were presented to reveal 'spread' of data.
Comment: Discussion - the reason why substitution of LS filler with GRP-WP causes changes to the properties of the asphalt concrete need to be presented and discussed (it is also unlikely that glass fibers in the GRP-WP influence 'strength' of mix as the grinding process will reduce their aspect ratio to much below the critical value for effective 'stiffening/strengthening'.
Response: The authors appreciate the reviewer’s comment. The discussion section was improved to explain why and how GRP-WP influences stiffness and volumetric properties of the mixes:
“In order to identify the reasons why GRP-WP filler replacement increases the performance of asphalt mixes and how addition of GRP-WP filler replacement into the mix changes the micro-structure and behavior of the mix, Scanning Electron Microscope (SEM) images of the asphalt mixes were also analyzed. It is known that GRP is a well-designed high-performance composite containing glass fiber, silica sand, and polyester resin in its structure. Having these materials in its structure, GRP is a very well-interlocked and tightly bonded material. GRP-WP contains considerable amount of micro-size chopped glass fibers (CGF), silica sand particles and polyester resin particles in its structure, when added into the asphalt mixes, they bond all particles in the asphalt mix together, significantly increasing performance of the mixes.“
The following sentences are newly added into the discussion section of the revised paper (lines 279-292):
“This explains why the samples with GRP-WP exhibit similar or better MS and flow values compared to the control samples. Moreover, there was a general trend among the samples with GRP-WP that as GRP-WP content increases, VMA and Va results increase but VFA results decrease. Effect of GRP-WP content on these volumetric properties are related to the amount, unit weight, particle shape, size, properties and grading of the GRP-WP. Having a lower unit weight compared to limestone, increasing GRP-WP percentage in the mixtures leads to less dense mixtures. Higher Va results with increasing GRP-WP content might be due to lower level of adsorption and chemical exchange between silica and asphalt, causing more free binder in the mixtures. It was also observed during the compaction of the mixtures that even though the same number and level of Marshall hammer blows were used in all samples, a noticeable decrease in the level of compaction occurred as GRP-WP content increases. This might be one of the main reasons why Va results increase as GRP-WP content increases. VFA is inversely related to Va: as Va increases, the VFA decreases. That is why, as GRP-WP content increases, Va results increase but VFA results decrease.”
Comment: Please refer to comments in attached annotated manuscript: Line 54 “Despite of being”
Response: The authors appreciate the reviewer’s comment. “Despite of being” was replaced with “Despite being” in the revised manuscript to correct the grammar mistake.
Comment: Line 56 - Please check all references as Choudhary et al is reference [13]
Response: The authors appreciate the reviewer’s comment. Choudhary et al. reference was relocated to be the ninth reference in the reference list, as pointed out by the reviewer. All other references were double-checked to ensure that their locations in the reference list are correct.
Comment: Line 83 - rice hush ash
Response: The authors appreciate the reviewer’s comment. The typo in “rice hush ash” was corrected as “rice husk ash” in the revised manuscript.
Comment: Lines 86-87 regarding “However, there is no study found in the literature investigating use of Glass fiber reinforced polyester (GRP) waste as filler in asphalt mixes.” Several articles found for use of GRP waste in construction concrete products (paving - see Osmani and Pappu (2010) http://www.claisse.info/2010%20papers/m45.pdf ) and please investigate US patent US20040132842A1 System and method for recycling scrap fiberglass products in concrete and asphalt construction
Response: The authors appreciate the reviewer’s comment. The first reference “Osmani and Pappu (2010)” was reviewed, and it was found that this reference is related to the use of fiber glass reinforced plastic waste in concrete and cement composites rather than in asphalt. In the literature review section of the paper, in lines 86-89 of the revised manuscript, the studies related to use of GRP in concrete-related mixes are summarized. Since the study “Osmani and Pappu (2010)” is related to the use of GRP in concrete, it was also cited in the revised manuscript with the reference number “41” in the reference list and mentioned in the text that it was one of the concrete-related studies that GRP was used, as requested by the reviewer:
“The reutilization of the fine GRP powder as a partial cement replacement, partial fine aggregate replacement, filler addition for self-compacting concrete, as well as its influence on the durability of the cementitious products have already been widely exploited [34-41].”
In terms of the second reference, as explained in the response to one of the previous comments, US patent (US 2004/0132842 A1) that reviewer mentioned was thoroughly reviewed. The patent was found to be only related to the methods (using what kind of techniques and machinery) that scrap fiberglass could be used in both concrete and asphalt construction. Although it is not directly related to the scope of the paper (the scope of the paper is finding the optimum GRP-WP content to be used as a filler in asphalt mix), the authors decided to cite this US patent in the literature review part of the revised manuscript, as suggested by the reviewer:
The sentence in lines 86-87 “However, there is no study found in the literature investigating use of Glass fiber reinforced polyester (GRP) waste as filler in asphalt mixes.” was revised in the revised manuscript as “ However, there are not many studies found in the literature investigating use of Glass fiber reinforced polyester (GRP) waste as filler in asphalt mixes [34].” Reference [34] is for the US patent.
Comment: Lines 110-112 appear to be added as an afterthought - this should have been introduced earlier - p2 line 86?
Response: The authors appreciate the reviewer’s comment. As per reviewer’s request, the sentence in lines 110-112 “The reutilization of the fine GRP powder as a partial cement replacement, partial fine aggregate replacement, filler addition for self-compacting concrete, as well as its influence on the durability of the cementitious products has been already widely exploited [34-41].” was relocated to line 86 in the revised menuscript.
Comment: Section 1 (introduction) could be condensed as there is some repetition and the 'presentation order' for literature could be improved - e.g. condense section on current research and use of GRP waste in construction products (asphalt/concrete/blocks etc)
Response: The authors appreciate the reviewer’s comment. As stated in the response to the previous comment, the sentence in lines 110-112 was relocated to line 86 to make the Introduction section more organized.
Comment: I suggest you include a reference for the Marshall stability test (ASTM D6927)
Response: The authors appreciate the reviewer’s comment. As per the reviewer’s request, a reference for the Marshall stability test was added in the revised manuscript.
Comment: Lines 165-166: What are the specification design limits? How does air content affect 'behaviour' of asphalt?
Response: The authors appreciate the reviewer’s comment. Specification design limits for the air content are presented in Figure 8. As shown in Figure 8, specification limits for the air content is 6-8%. It is a well-known fact that a lower and upper limit has to be specified for the air content as too great or low air contents can cause a significant reduction in pavement life. This is because, greater air content tends to cause higher level of rutting, raveling and moisture damage as well as lower level of stiffness and strength while insufficient air content makes the asphalt mix plastic and unstable, and causing flushing.
Comment: Line 182: I assume that the results are the average of 3 samples - you should either plot all values or at least give an indication of the range for each sample mix (SD or error)
Response: The authors appreciate the reviewer’s comment. As stated in lines 155-, “For each binder content, three samples were prepared, making a total of 15 samples (3x5). All the produced samples were tested based on Marshall stability (MS) test and stability, flow as well as bulk specific gravity (Gsb), air content (Va), voids in mineral aggregate (VMA) and voids filled with asphalt (VFA) results were obtained so that optimum binder content could be determined (Figure 3 a-f).” Results shown in the figures in Results section are based on the mean values of the three samples tested for each case. As requested by the reviewer, the figures were revised in the revised manuscript that both mean value and the ranges of the test results for each test were presented to reveal 'spread' of data.
Comment: Lines 208-209: It is strange that the control only just meets the flow criteria?
Response: The authors appreciate the reviewer’s comment. As stated in lines 209-210, along with the control samples, the samples with 4.5% and 5% binder contents and 3.75% GRP-WP and 1.25 % LS filler contents also satisfy the specification limits in terms of flow criterion. Control samples also meet the corresponding specification limits for MS stability, VMA, VFA and air content.
Comment: Lines 235-236: ?? needs better wording??
Response: The authors appreciate the reviewer’s comment. The sentence in lines 235-236 “It was observed that for all binder content cases, Va results increase as GRP-WP content increases, except for the control samples cases, reaching its maximum value for the samples with 5% GRP-WP and no LS filler contents.” was revised in the revised manuscript as "It was observed that Va results increase as GRP-WP content increases. However, Va results of the control samples were between that of the samples with 2.5% GRP-WP and 2.5% LS filler contents, and 3.75% GRP-WP and 1.25% LS filler contents.”
Comment: Lines 239-243: These observations are different from my interpretation from fig 8 - limits drawn are 4% to 6% and only 3.75% GRP-WP sample VA values fall between these limits!
Response: The authors appreciate the reviewer’s comment. The sentence in lines 238-243 “Samples with the Va results within the limits of the specification were as follows: all control samples, only the samples with 4% binder contents among the sample groups with 1.25% GRP-WP and 3.75% LS filler contents, and 2.5% GRP-WP and 2.5% LS filler contents, all samples with 3.75% GRP-WP and 1.25% LS filler contents. However, no samples with 5% GRP-WP and no LS filler contents were found to meet the Va requirements [49].” was revised as “Samples with the Va results within the limits of the specification were as follows: all control samples and all samples with 3.75% GRP-WP and 1.25% LS filler contents. However, no samples with 5% GRP-WP and no LS filler contents were found to meet the Va requirements [49].” In the revised manuscript.
Comment: Lines 270-278: This paragraph is important and should be part of a larger discussion section. For this article to be accepted significant discussion needs to be added to propose why the values measured (MS/flow/VMA/VFA/VA) change with changes in GRP_WP content.
Response: The authors appreciate the reviewer’s comment. An extensive discussion was added into the revised paper explaining why measured (MS/flow/VMA/VFA/VA) values change with -changes in GRP-WP content. Newly added discussion part is in lines 279-292 as:
“This explains why the samples with GRP-WP exhibit similar or better MS and flow values compared to the control samples. Moreover, there was a general trend among the samples with GRP-WP that as GRP-WP content increases, VMA and Va results increase but VFA results decrease. Effect of GRP-WP content on these volumetric properties are related to the amount, unit weight, particle shape, size, properties and grading of the GRP-WP. Having a lower unit weight compared to limestone, increasing GRP-WP percentage in the mixtures leads to less dense mixtures. Higher Va results with increasing GRP-WP content might be due to lower level of adsorption and chemical exchange between silica and asphalt, causing more free binder in the mixtures. It was also observed during the compaction of the mixtures that even though the same number and level of Marshall hammer blows were used in all samples, a noticeable decrease in the level of compaction occurred as GRP-WP content increases. This might be one of the main reasons why Va results increase as GRP-WP content increases. VFA is inversely related to Va: as Va increases, the VFA decreases. That is why, as GRP-WP content increases, Va results increase but VFA results decrease.”

Reviewer 2 Report
Interesting study on interesting topic, but analysis is wanting. Authors need to do a better job explaining why they are getting the results than just describing the numbers. I think the main driving factor of all you results is the different air content you get is the different air void content, but why did the filler replacement initially decrease the air voids but then increase them? This is what we need to know before your results are relevant for research and industry.
Mention total filler content in abstraction (5%?) and define LS somewhere
Rather than saying what the "optimum mixture" was in the abstract and conclusion, which is subjective to what the mixture parameters are, I think it would be better to say the trends in mixture performance with increased filler replacement. What are the benefits and limitations?
Abstract is written word for word in conclusion.
Please consider this work on filler replacement for literature review:
https://doi.org/10.1016/j.conbuildmat.2020.120166
Author Response
Manuscript ID: materials-935635
Title: Use of GRP Pipe Waste Powder as a Filler Replacement in Hot-Mix Asphalt
Authors: Ahmet BeycioÄŸlu, Orhan Kaya, Zeynel Baran Yıldırım, Baki BaÄŸrıaçık, Magdalena Dobiszewska*, Nihat Morova, Suna Çetin
The authors sincerely appreciate all the constructive comments received from the reviewers for this paper. All of the comments were thoroughly addressed in the revised version of the manuscript. Detailed responses to the reviewer comments are presented below.
REVIEWER 2:
Comment: Interesting study on interesting topic, but analysis is wanting. Authors need to do a better job explaining why they are getting the results than just describing the numbers. I think the main driving factor of all you results is the different air content you get is the different air void content, but why did the filler replacement initially decrease the air voids but then increase them? This is what we need to know before your results are relevant for research and industry.
Response: The authors appreciate the reviewer’s comment. As stated in the response to the last comment of the Review 1, an extensive discussion was added into the revised paper explaining why Va results along with the results of the other parameters change with changes in GRP-WP content. Newly added discussion part is in lines 279-292 as:
“This explains why the samples with GRP-WP exhibit similar or better MS and flow values compared to the control samples. Moreover, there was a general trend among the samples with GRP-WP that as GRP-WP content increases, VMA and Va results increase but VFA results decrease. Effect of GRP-WP content on these volumetric properties are related to the amount, unit weight, particle shape, size, properties and grading of the GRP-WP. Having a lower unit weight compared to limestone, increasing GRP-WP percentage in the mixtures leads to less dense mixtures. Higher Va results with increasing GRP-WP content might be due to lower level of adsorption and chemical exchange between silica and asphalt, causing more free binder in the mixtures. It was also observed during the compaction of the mixtures that even though the same number and level of Marshall hammer blows were used in all samples, a noticeable decrease in the level of compaction occurred as GRP-WP content increases. This might be one of the main reasons why Va results increase as GRP-WP content increases. VFA is inversely related to Va: as Va increases, the VFA decreases. That is why, as GRP-WP content increases, Va results increase but VFA results decrease.”
Comment: Mention total filler content in abstraction (5%?) and define LS somewhere
Response: The authors appreciate the reviewer’s comment. As suggested by the reviewer, the statement in the Abstract section “45 samples with three different binder contents (4%, 4.5% and 5.0%) and five different GRP-WP contents (0%, 25%, 50%, 75% and 100% replacement by weight of the filler)” was replaced with “45 samples with three different binder contents (4%, 4.5% and 5.0%), and a 5% filler content with five different percentages of the GRP-WP contents (0%, 25%, 50%, 75% and 100% replacement by weight of the filler)” in revised manuscript to mention that filler content was 5% for all the samples. Also, the acronym “LS” was replaced with the full word “limestone” in the Abstract section and “LS” acronym was defined in the body part of the revised manuscript.
Comment: Rather than saying what the "optimum mixture" was in the abstract and conclusion, which is subjective to what the mixture parameters are, I think it would be better to say the trends in mixture performance with increased filler replacement. What are the benefits and limitations?
Response: The authors appreciate the reviewer’s comment. “Optimum mixture” was defined in lines 173-174 as “the mixtures producing the best Marshall stability and flow values as well as satisfying specification limits. All Results section starting from page 3 explains the trends in mixture performance with increased filler replacement such as: line (214) ”VMA results were found to be increasing as GRP-WP content increases”, (line 224) VFA results tend to decrease as GRP-WP content increases.”(line 233) “Va results tend to increase as GRP-WP content increases”.
As stated to the response to the previous comments, a new part was added to discussion part describing benefits and limitations of GRP-WP. In terms of the benefits, it was found that (line 280) “the samples with GRP-WP exhibit similar or better MS and flow values compared to the control samples”. In terms of limitations, it was mentioned in the discussion part that (lines 287-289) “It was also observed during the compaction of the mixes that even though the same number and level of Marshall hammer blows were used in all samples, a noticeable decrease in the level of compaction occurred as GRP-WP content increases”. This sentence explains that in order to successfully use GRP-WP as a filler, the samples should be well-compacted to avoid excessive air voids.
Comment: Abstract is written word for word in conclusion.
Response: The authors appreciate the reviewer’s comment. In conclusion section, findings of this study were presented in detail and a through discussion was added. As explained in the previous comments, discussion portion of the Discussion and Conclusion section was revised to add more discussion.
Comment: Please consider this work on filler replacement for literature review:
https://doi.org/10.1016/j.conbuildmat.2020.120166
Response: The authors appreciate the reviewer’s comment. This reference was added into the revised manuscript with a reference number “49”, as suggested by the reviewer.

Round 2
Reviewer 1 Report
The discussion regarding the importance of the filler content in asphalt concrete and how replacing limestone filler with GRP waste filler affects 'performance' could be strengthened.
Author Response
Manuscript ID: materials-935635
Title: Use of GRP Pipe Waste Powder as a Filler Replacement in Hot-Mix Asphalt
Authors: Ahmet BeycioÄŸlu, Orhan Kaya, Zeynel Baran Yıldırım, Baki BaÄŸrıaçık, Magdalena Dobiszewska*, Nihat Morova, Suna Çetin
The authors sincerely appreciate all the constructive comments received from the reviewers for this paper. All of the comments were thoroughly addressed in the revised version of the manuscript. Detailed responses to the reviewer comments are presented below.
REVIEWER 1:
Comment: The discussion regarding the importance of the filler content in asphalt concrete and how replacing limestone filler with GRP waste filler affects 'performance' could be strengthened.
Response: The authors appreciate the reviewer’s comment. Based on the reviewer’s comment, Discussion and Conclusion section of the paper was extensively revised in the revised manuscript to explain the importance of the filler content in asphalt concrete and how replacing limestone filler with GRP waste filler affects performance.
Regarding “importance of the filler content in asphalt concrete”, the following statements were added/revised in the revised manuscript (lines 282-289):
“It is known that when mixed with asphalt binder, filler and asphalt binder creates a filler-asphalt mastic, a high-consistency matrix, cementing larger aggregate particles together, and so affecting mechanical and physical properties of the asphalt mixtures [52]. GRP-WP contains considerable amount of micro-size chopped glass fibers (CGF), silica sand particles and polyester resin particles in its structure. When it is mixed with asphalt binders, due to its fibrous properties, it might have produced a stiff mastic that binds aggregate particles together and produce a mixture with a similar performance as the mixtures with LS fillers. This might explain why the samples with GRP-WP exhibited similar MS and flow values compared to the control samples.”
Regarding “how replacing limestone filler with GRP waste filler affects performance”, the detailed explanation on how GRP-WP affected the performance was provided in the following sentences that were added/revised in the revised manuscript (lines 262-289):
“One of the ways to evaluate if a mixture provides similar performance is to compare MS and flow results of the mixtures with the control samples. MS is measure of strength in asphalt mixtures. Higher values of MS results of the asphalt mixtures are desirable in order to show them they resist shoving and rutting [51]. Therefore, a minimum MS value is specified for the asphalt mixtures, considering the level of traffic the mixture designed for. In terms of flow results, the maximum allowable flow values in the specifications control the plasticity and maximum allowable binder contents while the lowest flow values control brittleness and strength of the mixes [51]. Therefore, it is required that the flow results of the asphalt mixtures must be between the lower and upper specification limits. Overall, it was found that the samples prepared with GRP-WP filler replacement produced similar MS and flow values compared to the control samples as well as satisfying the specification limits, except for the case when all LS fillers were replaced with GRP-WP. The samples with 5% GRP-WP and no LS fillers produced consistently poorer performance compared to the control samples. One of the main objectives of this study was satisfied that there is not significant loss in the performance of the samples with GRP-WP, except for the case with 5% GRP-WP and no LS fillers, compared to the control samples.
In order to identify how GRP-WP filler replacement changes the micro-structure and behavior of the mix, Scanning Electron Microscope (SEM) images of the asphalt mixes were also analyzed. It is known that GRP is a well-designed high-performance composite containing glass fiber, silica sand, and polyester resin in its structure [48]. Having these materials in its structure, GRP is a very well-interlocked and tightly bonded material. It is known that when mixed with asphalt binder, filler and asphalt binder creates a filler-asphalt mastic, a high-consistency matrix, cementing larger aggregate particles together, and so affecting mechanical and physical properties of the asphalt mixtures [52]. GRP-WP contains considerable amount of micro-size chopped glass fibers (CGF), silica sand particles and polyester resin particles in its structure. When it is mixed with asphalt binders, due to its fibrous properties, it might have produced a stiff mastic that binds aggregate particles together and produce a mixture with a similar performance as the mixtures with LS fillers. This might explain why the samples with GRP-WP exhibited similar MS and flow values compared to the control samples.“

Reviewer 2 Report
While the efforts of the authors to improve the results are not, the analysis remains on the technical level of describing the trends rather than an actual scientific analysis. There are three main scientific statements found in the discussion, none of which contain any references backing them up:
"Having these materials in its structure, GRP is a very well-interlocked and tightly bonded material. GRP-WP contains considerable amount of micro-size chopped glass fibers (CGF), silica sand particles and polyester resin particles in its structure, when added into the asphalt mixes, they bond all particles in the asphalt mix together, significantly increasing performance of the mixes."
- Are you saying the GRP-WP serves as a binder? This is surprising since it is a mineral. Please clarify.
"Effect of GRP-WP content on these volumetric properties are related to the amount, unit weight, particle shape, size, properties and grading of the GRP-WP. Having a lower unit weight compared to limestone, increasing GRP-WP percentage in the mixtures leads to less dense mixtures."
- This is a very vague statement and does not provide any clarity to the reader
"Higher Va results with increasing GRP-WP content might be due to lower level of adsorption and chemical exchange between silica and asphalt, causing more free binder in the mixtures."
- This is a hypothesis but with no back up in terms of references
Finally, only the Va and Vfa results have any explanations, while not relating them to performance. The more important results of flow and stability are not related.
You also keep on using biased terms like "best" to describe these results, without explaining how the "best" flow and stability is better for the road.
Author Response
Manuscript ID: materials-935635
Title: Use of GRP Pipe Waste Powder as a Filler Replacement in Hot-Mix Asphalt
Authors: Ahmet BeycioÄŸlu, Orhan Kaya, Zeynel Baran Yıldırım, Baki BaÄŸrıaçık, Magdalena Dobiszewska*, Nihat Morova, Suna Çetin
The authors sincerely appreciate all the constructive comments received from the reviewers for this paper. All of the comments were thoroughly addressed in the revised version of the manuscript. Detailed responses to the reviewer comments are presented below.
REVIEWER 2:
Comment: While the efforts of the authors to improve the results are not, the analysis remains on the technical level of describing the trends rather than an actual scientific analysis. There are three main scientific statements found in the discussion, none of which contain any references backing them up:
"Having these materials in its structure, GRP is a very well-interlocked and tightly bonded material. GRP-WP contains considerable amount of micro-size chopped glass fibers (CGF), silica sand particles and polyester resin particles in its structure, when added into the asphalt mixes, they bond all particles in the asphalt mix together, significantly increasing performance of the mixes."
Are you saying the GRP-WP serves as a binder? This is surprising since it is a mineral. Please clarify.
Response: The authors appreciate the reviewer’s comment. To clarify this sentence, sentences in lines 274-277 “In order to identify the reasons why GRP-WP filler replacement increases the performance of asphalt mixes and how addition of GRP-WP filler replacement into the mix changes the micro-structure and behavior of the mix, Scanning Electron Microscope (SEM) images of the asphalt mixes were also analyzed. It is known that GRP is a well-designed high-performance composite containing glass fiber, silica sand, and polyester resin in its structure. Having these materials in its structure, GRP is a very well-interlocked and tightly bonded material. GRP-WP contains considerable amount of micro-size chopped glass fibers (CGF), silica sand particles and polyester resin particles in its structure, when added into the asphalt mixes, they bond all particles in the asphalt mix together, significantly increasing performance of the mixes. This explains why the samples with GRP-WP exhibit similar or better MS and flow values compared to the control samples.” were replaced with the sentences “In order to identify how GRP-WP filler replacement changes the micro-structure and behavior of the mix, Scanning Electron Microscope (SEM) images of the asphalt mixes were also analyzed. It is known that GRP is a well-designed high-performance composite containing glass fiber, silica sand, and polyester resin in its structure [48]. Having these materials in its structure, GRP is a very well-interlocked and tightly bonded material. It is known that when mixed with asphalt binder, filler and asphalt binder creates a filler-asphalt mastic, a high-consistency matrix, cementing larger aggregate particles together, and so affecting mechanical and physical properties of the asphalt mixtures [52]. GRP-WP contains considerable amount of micro-size chopped glass fibers (CGF), silica sand particles and polyester resin particles in its structure. When it is mixed with asphalt binders, due to its fibrous properties, it might have produced a stiff mastic that binds aggregate particles together and produce a mixture with a similar performance as the mixtures with LS fillers. This might explain why the samples with GRP-WP exhibited similar MS and flow values compared to the control samples.” in the revised manuscript to clarify that “filler-asphalt mastic” binds large aggregates together, not the filler.
Also, two references (48 and 52) were added to the revised sentences to support the claims, as suggested by the reviewer.
Comment: "Effect of GRP-WP content on these volumetric properties are related to the amount, unit weight, particle shape, size, properties and grading of the GRP-WP. Having a lower unit weight compared to limestone, increasing GRP-WP percentage in the mixtures leads to less dense mixtures."
This is a very vague statement and does not provide any clarity to the reader.
Response: The authors appreciate the reviewer’s comment. These two sentences were deleted in the revised paper to eliminate any confusion, as suggested by the reviewer.
Comment: "Higher Va results with increasing GRP-WP content might be due to lower level of adsorption and chemical exchange between silica and asphalt, causing more free binder in the mixtures."
This is a hypothesis but with no back up in terms of references
Response: The authors appreciate the reviewer’s comment. A new reference (55) was added to support this hypothesis, (page 2 of the mentioned reference).
Comment: Finally, only the Va and Vfa results have any explanations, while not relating them to performance. The more important results of flow and stability are not related.
Response: The authors appreciate the reviewer’s comment. Four new paragraphs were added into the Discussion and Conclusion section of the revised manuscript (lines 290-317) to explain the relationships between volumetric properties of the mixtures and their performance. Results of the flow and MS and their relationships with the volumetric properties were also discussed in the newly-added paragraphs. VMA results were also discussed along with Va and VFA results in the revised manuscript. The discussions were also supported by the newly added references in the revised manuscript [53-56]:
“Numerous studies showed that there are correlations between volumetric properties of the mixes and their performances [53, 54]. Durability of an asphalt mixture is highly affected by its Va. Too low Va may lead to flushing whereas too high Va might cause water damage and rutting in the mixtures. On the other hand, VMA shows the amount of space to accommodate the effective volume of binder and the volume of air voids needed in the asphalt mixture. In order to achieve a durable binder film thickness, a minimum VMA value is required [53]. VFA, the total amount of voids between aggregate particles in the compacted mixture that are filled with asphalt binder, is specified to avoid less durable asphalt mixes. It was also shown that a decrease in Va and increase in VFA are some of the ways to reduce cracking potential of the mixtures [54]. In order to produce a well-performing asphalt mixture, all volumetric properties of the mixes must satisfy corresponding specification limits.
There was a general trend among the samples with GRP-WP that as GRP-WP content increases, VMA and Va results increase but VFA results decrease. Higher Va results with increasing GRP-WP content might be due to lower level of adsorption and chemical exchange between silica and asphalt, causing more free binder in the mixtures [55]. It was also observed during the compaction of the mixtures that even though the same number and level of Marshall hammer blows were used in all samples, a noticeable decrease in the level of compaction occurred as GRP-WP content increases. This might be one of the main reasons why Va results increase as GRP-WP content increases. A similar trend was observed in some other studies that [56, 57] reduction in compaction might cause an increase in Va and VMA and a reduction in VFA.
It was observed in this study that the samples with 5% GRP-WP contents with no LS filler exhibited significantly higher level of Va and VMA as well as significantly lower level of VFA compared to the control samples. As a result, the same samples produced significantly lower MS and flow results compared to that of the control samples. It shows that the volumetric properties of the samples are one of the keys factors in determining the performance of asphalt mixtures.
Volumetric properties of asphalt mixtures are related to each other. VFA is inversely related to Va and VMA: as Va and VMA increase, the VFA decreases. That is why, as GRP-WP content increases, Va and VMA results increase but VFA results decrease [56].”
Comment: You also keep on using biased terms like "best" to describe these results, without explaining how the "best" flow and stability is better for the road.
Response: The authors appreciate the reviewer’s comment. The term “best” was deleted throughout the revised paper, and more clear statements such as “the highest Marshall stability as well as satisfying specification limits for flow” were added instead of the statement “best MS and flow”. Also, why stability and flow values are important for the road was explained in the newly-added sentences in the revised manuscript (lines 263-271):
“Higher values of MS results of the asphalt mixtures are desirable in order to show them they resist shoving and rutting [51]. Therefore, a minimum MS value is specified for the asphalt mixtures, considering the level of traffic the mixture designed for. In terms of flow results, the maximum allowable flow values in the specifications control the plasticity and maximum allowable binder contents while the lowest flow values control brittleness and strength of the mixes [51]. Therefore, it is required that the flow results of the asphalt mixtures must be between the lower and upper specification limits.”

Round 3
Reviewer 2 Report
There were significant improvements made to the analysis, but I think it is critical here to explain not just how a filler works, but why this filler performs differently from the control. Additionally, it is evident that the MS results are almost entirely tied to the air voids content variation in the samples, which the authors do not point out. The MS of a higher air void sample will be lower: this is nothing new.
What should be explained is why the compaction is initially easier with small replacement of the filler, but more difficult with higher replacement rates. If this cannot be explained, then I do not see this as more than a conference paper or technical report.
Author Response
Manuscript ID: materials-935635
Title: Use of GRP Pipe Waste Powder as a Filler Replacement in Hot-Mix Asphalt
Authors: Ahmet BeycioÄŸlu, Orhan Kaya, Zeynel Baran Yıldırım, Baki BaÄŸrıaçık, Magdalena Dobiszewska*, Nihat Morova, Suna Çetin
The authors sincerely appreciate all the constructive comments received from the reviewers for this paper. All of the comments were thoroughly addressed in the revised version of the manuscript. Detailed responses to the reviewer comments are presented below.
REVIEWER:
Comment: There were significant improvements made to the analysis, but I think it is critical here to explain not just how a filler works, but why this filler performs differently from the control. Additionally, it is evident that the MS results are almost entirely tied to the air voids content variation in the samples, which the authors do not point out. The MS of a higher air void sample will be lower: this is nothing new.
Response: The authors appreciate the reviewer’s comment. In response to the comment “why this filler performs differently from the control.”, the following two paragraphs were revised in the revised manuscript (lines 301-324) to address why this filler performs differently from the control:
“There was a general trend among the samples with GRP-WP that as GRP-WP content increases, VMA and Va results increase but VFA results decrease. Higher Va results with increasing GRP-WP content might be due to lower level of adsorption and chemical exchange between silica and asphalt, causing more free binder in the mixtures [55]. In other words, as found by some other studies [56], GRP-WP has lower porosity compared to LS. Lower porosity of GRP-WP causes lower absorbance of bitumen in the mixture leading to more free binder in the mixtures. Therefore, as GRP-WP filler content increases, the amount of free binder in the mixture increases, causing higher level of Va of the mixes [55, 56]. In order to solve that issue, an optimum asphalt binder content could be determined for each GRP-WP filler content to avoid excessive Va.
It was also observed during the compaction of the mixtures that even though the same number and level of Marshall hammer blows were used in all samples, a noticeable decrease in the level of compaction occurred as GRP-WP content increases. Zulkati et al. [13] demonstrated that stiffened asphalt-filler mastic produces a tougher mix that is hard to compact. The role of filler in the asphalt-binder mastic is that fillers act as tiny rollers during the compaction, similar to the role of friction-lubricate agents, given that they have regular and spherical shapes. The asphalt-filler mastic with regular-shaped fillers, such as LS [13], leads to a faster and smoother orientation of the larger aggregates, causing a less compaction resistance. As discussed earlier based on Figure 2 that the SEM image of the GRP-WP reveals that GRP-WP contains considerable amount of micro-size chopped glass fibers (CGF). Due to the irregular shapes of these CGFs, they have significantly longer lengths compared to their widths, they might have negatively impacted the workability of the asphalt mixtures causing a reduction in compaction, as discussed by several other studies presented by Melotti et al. [57] and Wróbel et al. [58]. Resistance to compaction might be one of the main reasons why Va results increase as GRP-WP content increases. A similar trend was observed in some other studies that [59, 60] reduction in compaction might cause an increase in Va.”
In response to the comment “Additionally, it is evident that the MS results are almost entirely tied to the air voids content variation in the samples, which the authors do not point out”, the following paragraph (lines 325-329) addresses this comment:
“It was observed in this study that the samples with 5% GRP-WP contents with no LS filler exhibited significantly higher level of Va and VMA as well as significantly lower level of VFA compared to the control samples. As a result, the same samples produced significantly lower MS and flow results compared to that of the control samples. It shows that the volumetric properties of the samples are one of the keys factors in determining the performance of asphalt mixtures.”
Comment: What should be explained is why the compaction is initially easier with small replacement of the filler, but more difficult with higher replacement rates. If this cannot be explained, then I do not see this as more than a conference paper or technical report.
Response: The authors appreciate the reviewer’s comment. the following paragraph was revised in the revised manuscript (lines 310-324) to address why the compaction is initially easier with small replacement of the filler, but more difficult with higher replacement rates:
“It was also observed during the compaction of the mixtures that even though the same number and level of Marshall hammer blows were used in all samples, a noticeable decrease in the level of compaction occurred as GRP-WP content increases. Zulkati et al. [13] demonstrated that stiffened asphalt-filler mastic produces a tougher mix that is hard to compact. The role of filler in the asphalt-binder mastic is that fillers act as tiny rollers during the compaction, similar to the role of friction-lubricate agents, given that they have regular and spherical shapes. The asphalt-filler mastic with regular-shaped fillers, such as LS [13], leads to a faster and smoother orientation of the larger aggregates, causing a less compaction resistance. As discussed earlier based on Figure 2 that the SEM image of the GRP-WP reveals that GRP-WP contains considerable amount of micro-size chopped glass fibers (CGF). Due to the irregular shapes of these CGFs, they have significantly longer lengths compared to their widths, they might have negatively impacted the workability of the asphalt mixtures causing a reduction in compaction, as discussed by several other studies presented by Melotti et al. [57] and Wróbel et al. [58]. Resistance to compaction might be one of the main reasons why Va results increase as GRP-WP content increases. A similar trend was observed in some other studies that [59, 60] reduction in compaction might cause an increase in Va.”
